# Association between Perceived Oral Symptoms and Presence of Clinically Diagnosed Oral Diseases in a Sample of Pregnant Women in Malaysia

**DOI:** 10.3390/ijerph17197337

**Published:** 2020-10-08

**Authors:** Ema Yunita Sari, Norkhafizah Saddki, Azizah Yusoff

**Affiliations:** School of Dental Sciences, Health Campus, Universiti Sains Malaysia, Kubang Kerian 16150, Kelantan, Malaysia; ema_y_sari@yahoo.com (E.Y.S.); drazizah@usm.my (A.Y.)

**Keywords:** pregnant women, oral health, dental caries, periodontal diseases, prenatal care

## Abstract

The integration of oral health into primary health care denotes the important role of medical counterparts as the front liners in antenatal care to help screen mothers for oral symptoms and refer them to dentists accordingly. However, the validity of self-perceived oral health status is inconclusive. This study determined the association between self-perceived oral symptoms and the presence of clinically diagnosed oral diseases in a sample of pregnant women. A total of 192 pregnant women participated in this cross-sectional study. Clinical oral examinations were performed to record dental caries experience, gingival health and periodontal health. The women were also asked about their oral symptoms. Most women had at least one oral symptom (84.9%): cavitated tooth (62.0%), bad breath (38.5%), bleeding gums (28.6%), and toothache (22.9%). About half of the women had untreated dental caries (58.9%), and the odds were significantly higher in women who complained of having cavitated tooth. About half of the women had moderate to severe gingivitis (53.7%), and the odds were significantly higher in women who complained of bleeding gums. About half had periodontal pockets (46.3%), and the odds were higher in women who complained of bleeding gums and bad breath although lower in women who complained of swollen gums. In conclusion, the prevalence of dental caries and periodontal disease among pregnant women in this study were relatively high. The presence of untreated dental caries, moderate and severe gingivitis, and periodontal pockets were significantly associated with their corresponding oral symptoms.

## 1. Introduction

Pregnancy is associated with various physiological, anatomical, and hormonal changes that affect not only the reproductive system, but also other organ systems of the body including the oral cavity. While high levels of progesterone and oestrogen are essential for a healthy and successful pregnancy, their effects on periodontal tissues are often detrimental [1]. These include changes in gingival vasculature, immune responses, and connective tissue metabolism which have been linked to increased susceptibility of gingival tissues to dental plaque [1]. 

Hormonal changes during pregnancy have also been implicated as the most likely factor for morning sickness [2]. The experience of nausea and vomiting may not only deter routine oral hygiene practices, but also lead to frequent consumption of acidogenic meals and snacks to ward off nausea, increasing the women’s risk for dental caries [3]. Additionally, gastric acid produced by vomiting can erode enamel surfaces, making the teeth more sensitive and susceptible to dental caries. The prevalence of dental caries in pregnant women has been reported to be high in studies among different populations [4,5,6].

Despite being at an increased risk for oral diseases, the uptake of oral health care services among pregnant women has been relatively low and most women visited dentists only when they had symptoms like pain [7,8]. While visits to dental clinics have been low, about 86% of pregnant women worldwide attended antenatal care at least once during their pregnancy [9]. These statistics indicate the potential role of medical care providers in preventive oral health care. The regularity and frequency of contact with doctors, nurses, or midwives during antenatal care provide an opportunity for the medical counterparts to screen women for oral health problems and refer them to dentists accordingly. In medicine, self-report is an accepted diagnostic tool for certain clinical conditions such as the Epworth Sleepiness Scale for obstructive sleep apnoea, the Health Assessment Questionnaire for rheumatic diseases, and the Fagerström Test for nicotine dependence. Nevertheless, there is inconclusive evidence in the literature regarding the validity of self-perceived oral health status [10,11,12,13].

This study aimed to determine the association between self-perceived oral health problems and professionally diagnosed oral health status in pregnant women.

## 2. Materials and Methods 

### 2.1. Population and Sample

This was a cross-sectional study of pregnant women who received antenatal care from the Obstetrics and Gynaecology Specialist Clinic at Hospital Universiti Sains Malaysia (USM), Kelantan. Hospital USM is teaching hospital and tertiary referral center. Complementing the role of the Ministry of Health Malaysia to preserve the health of the people, the basic medical and health care services including antenatal care are provided without charge to the public. 

Sample size for the study was determined using the formula for estimating a single proportion with requirement of 95% confidence [14]. The proportion of pregnant women with periodontal disease was estimated at 62% [4]. Sample sizes were calculated for various levels of precision, and a sample size of 185 with a precision of 0.07 (7%) was selected while taking into consideration available resources. To accommodate for about a 35% nonresponse rate, a sample size of 192 was finally decided. Potential participants were selected using a systematic random sampling method. Pregnant women with diabetes mellitus were excluded since they would be at higher risk of developing periodontal disease [1]. Pregnant women with acquired endocardial defects were also excluded due to risk of bacteraemia from dental probing [15]. This study received ethical approval from the USM Human Research and Ethics Committee (USMKK/PP/JEPeM[234.3(04)]). Written informed consent was obtained from all study participants prior to data collection.

### 2.2. Variables, Research Tools, and Data Collection

A structured self-administered questionnaire was used to obtain information on the women’s socio-demographic profile (age, ethnic group, educational level, employment status, and household income), obstetric profile (gestational period, gravida status, and parity status), and perceived oral health problems. Questions on socio-demographic and obstetric profile were short and straightforward with close-ended responses except the question on household income which was open-ended. A close-ended question was also used to ask the women if they ever experience any oral health problem during the current pregnancy. A fixed list of oral health problems was given with a response option of “yes” or “no” for each problem. The list was developed by a panel of clinical dental specialists. Additionally, we incorporated an open question through an “other” answer option with text entry on it for participants to specify other oral health problem they may have.

Clinical examination was performed to assess dental caries experience, gingival health, and periodontal health. Dental caries experience was measured using the Decayed, Missing, and Filled Teeth (DMFT) Index [16]. Third molars were excluded from the examination. The number of decayed (D), missing (M), and filled (F) permanent teeth (T) were recorded. The sum of D, M, and F will give the individual DMFT score.

The Gingival Index (GI) by Löe and Silness [17] was used to assess the severity of gingival inflammation. Four gingival areas of 6 index teeth (16, 12, 24, 36, 32 and 44) were examined. Missing teeth were not substituted. The following scores were given: 0 = normal healthy gingiva; 1 = mild inflammation with slight color change and oedema but no bleeding on probing; 2 = moderate inflammation with red, oedematous and glazing gingival tissues that bleed on probing; and 3 = severely inflamed gingival tissues with marked redness and oedema, ulceration, and tendency to bleed spontaneously. Scores from each index tooth were added and divided by 4 to give the GI for the tooth. The GI of each patient was determined by adding the scores for all index teeth and dividing it by the number of teeth examined. Gingival health was interpreted as follows: 0 = healthy; 0.1–1.0 = mild inflammation; 1.1–2.0 = moderate inflammation; and 2.1–3.0 = severe inflammation.

Periodontal health status was measured using the Community Periodontal Index (CPI) [16]. The mouth was divided into 6 sextants (18–14, 13–23, 24–28, 38-34, 33–43, 44–8) with 10 index teeth (17, 16, 11, 26, 27, 36, 37, 31, 47, and 46). If the index tooth is missing, all other teeth in the sextant would be examined. Each index tooth was probed using a lightweight CPI probe. The following scoring criteria were used: 0 = healthy periodontium, 1 = bleeding after probing, 2 = calculus detected during probing but the black band on the probe was visible, 3 = shallow pocket 4–5 mm (gingival margin within the black band on the probe), and 4 = deep pocket ≥ 6 mm (black band on the probe not visible). The highest CPI score of each patient was recorded, and the mean number of sextants having CPI code of 0, 1, 2, 3, and 4 were also calculated to indicate the severity of periodontal disease. 

### 2.3. Statistical Analysis

Data entry and analysis was done using the IBM SPSS statistics version 24.0 (IBM Corp., Armonk, NY, USA) Descriptive statistics, mean and standard deviation (SD) for continuous variables, and frequency and percentage (%) for categorical variables were determined. Simple logistic regression and multiple logistic regression analyses were done to determine the association between self-perceived oral health problems and the prevalence of untreated dental caries (DT), moderate and severe gingivitis (GI > 1.0), and periodontal pockets (CPI ≥ 3). The significance level was set at 0.05. Other independent variables were age, educational level, household income, employment status, period of gestation, gravida status, and parity status.

Following simple logistic regression analysis, variables with a *p*-value of less than 0.25 and variables that were deemed clinically relevant were included in the subsequent multivariable analysis. In multiple logistic regression analysis, variables were selected using the backward elimination Likelihood Ratio (LR) method. Following variable selection, all possible 2-way interactions and multicollinearity problems were checked. The overall model fitness was assessed using the Hosmer–Lemeshow goodness-of-fit test. Additionally, the predictive power of the model was assessed by checking its sensitivity and specificity, and area under the Receiver Operating Characteristic (ROC) curve. Influential outliers were identified using the Cook’s distance measure. Any observation with Cook’s distance greater than 1.0 is considered an influential outlier.

## 3. Results

All 192 selected women accepted to participate, giving a 100% participation rate. Socio-economic and obstetric profile of the participants are shown in Table 1. The age of the women ranged from 17 to 42 years with a mean age of 30.0 years (SD 5.37). Most of them were income earners and contributed to the household income that showed a positively skewed distribution. The median household income was Malaysian Ringgit (MYR) 3000 (interquartile range (IQR) 2500).

Most women reported having at least one oral health problem (84.9%). The four most common problems were cavitated tooth (62.0%), bad breath (38.5%), bleeding gums (28.6%), and toothache (22.9%). Other problems were gum pain (13.5%), swollen gums (10.9%), loose tooth (5.7%), and gum abscess (1.6%).

Table 2 shows dental caries experience, gingival health and periodontal health of the women. The prevalence of dental caries was 93.2% (95% CI: 89.6, 97.0) with a mean DMFT of 5.9 (95% CI: 5.31, 6.45). About half of them have untreated dental caries (58.9%, 95% CI: 51.83, 65.88) with a mean DT of 1.5 (95% CI: 1.20, 1.70). The prevalence of gingivitis was 100%, and about half have moderate to severe gingivitis (53.7%, 95% CI: 46.53, 60.76). Almost half of the women have periodontal pockets (46.3%, 95% CI: 39.24, 53.47) and only 1.0% had healthy periodontium.

The women’s education level and their perception of having cavitated tooth were significantly associated with prevalence of untreated dental caries (Table 3). While higher education levels have a protective effect against having untreated dental caries, the odds were higher in women who reported having cavitated tooth. Possible two-way interaction between the factors was not significant, and there was no multicollinearity problem. The Hosmer–Lemeshow goodness-of-fit test suggested that the model fits well (*p* = 0.867). The sensitivity and specificity of this model were 85.8% and 48.1%, respectively, and the overall correct classification result was 70.3%. The positive and negative predictive values were 81.9% and 70.4%, respectively. The area under the ROC curve was 75.1%. The contribution of each outlier was checked, and none was influential

Household income and perceived bleeding gums were significantly associated with prevalence of moderate to severe gingivitis (Table 4). Women with higher household income were less likely to have moderate to severe gingivitis. On the other hand, women with bleeding gums have higher odds of having moderate to severe gingivitis. There was no interaction or correlation between the independent variables. The model also fits well with sensitivity and specificity of 81.6% and 53.9%, respectively. The overall correct classification was 68.8% and the area under the ROC curve was 67.7%. The positive and negative predictive values were 67.2% and 71.6%, respectively. No influential outlier was detected

Table 5 shows that the period of gestation, perceived swollen gums, perceived bleeding gums, and perceived bad breath were significantly associated with prevalence of periodontal pockets. Women in the third trimester of pregnancy have higher odds of having periodontal pockets than those in the first or second trimester. While the odds of having periodontal pockets were higher for women who complained of bleeding gums and bad breath, the odds were lower for women with perceived swollen gums. Possible two-way interactions were not significant, and there was no multicollinearity problem. The model fits well with Hosmer–Lemeshow goodness-of-fit test *p*-value of 0.633. The sensitivity and specificity of this model was 41.6% and 75.7%, respectively, with an overall correct classification of 59.9%. The positive and negative predictive values were 59.7% and 60.0%, respectively. The area under the ROC curve was 63.1%. There was no influential outlier

## 4. Discussion

This study demonstrates the association between presence of oral symptoms and the corresponding oral diseases among pregnant women. Untreated oral diseases during pregnancy may not only cause pain and discomfort in women, but also increase their risk of delivering low birth weight and preterm babies [18]. Dental visit during pregnancy is therefore recommended, and many relevant authorities, including the Ministry of Health Malaysia, have produced guidelines for promoting and delivering oral health care during the antenatal period [19]. However, only 47.0% of pregnant women in Malaysia utilized primary oral health care services [20], compared to 94.5% who attended health clinics for antenatal care [21]. Our findings underlined the potential roles of medical care providers as the front liners in antenatal care to screen for symptoms of dental caries and periodontal diseases and refer the women to dentists accordingly.

Caries prevalence among pregnant women in this study was higher than the adult female population of Malaysia which was 89.8% [22]. Prevalence of dental caries was also high among pregnant women in other studies among different populations. A study in Manaus, Brazil reported a prevalence of 100% [4]. A high prevalence of 99.9% was also reported in a large study of 1070 pregnant women in Kaunas, Lithuania [6]. Studies in Manipal, India and Chiang Mai, Thailand reported caries prevalence of 84% and 74.5%, respectively [5,23]. The study in Chiang Mai also showed that dental caries prevalence was higher among pregnant women who were 2.9 times more likely to suffer from the disease than non-pregnant women [5]. While the risk may be attributed to decreased attention to oral hygiene and frequent intake of sugary foods and drinks to alleviate pregnancy cravings and to ward off nausea [3], development of dental caries usually takes several years, indicating that initiation and progression of the lesion may have started long before pregnancy. Thus, pregnant women’s caries experience may not be due to the pregnancy per se but associated with other determinants that have taken their toll prior to pregnancy such as socio-economic status, general health status, and lifestyle behavior such as oral hygiene practice, dietary pattern, and dental visit.

The mean DMFT of women in this study was 5.9 (SD 3.97). Lower mean DMFT, 4.08 (SD 3.6), was reported among pregnant women in South India [23]. Higher mean DMFT of 12.1 was reported among Lithuanian pregnant women [6]. A study in Argentina also reported a high mean DMFT of 12.24 [24], and separate studies in Brazil by Bressane et al. [4] and Tonello et al. [25] reported mean DMFT of 10.0 and 11.1, respectively. The large variation in dental caries experience among pregnant women between studies might be due to differences in composition of study samples. This variation also corresponds to the pattern of dental caries severity that differs between areas, countries and regions of the world, attributable to distinct living conditions, lifestyles and environmental factors, and the implementation of preventive oral health care programs [26].

A bit more than half of women in this study had untreated dental caries. This finding is comparable to the prevalence reported among pregnant women in France which was 51.6% [27]. A much higher prevalence was reported in Buenos Aires, Argentina; 92.1% with a mean DT of 6.46 [24]. A study among low-income Hispanic pregnant women at the California-Mexico border also reported a high prevalence of 93.4% [28].

The prevalence of untreated dental caries among pregnant women in this study was associated with the perception of having cavitated tooth, the most common oral symptom reported by the women. Another factor found to be associated with prevalence of untreated dental caries was the women’s level of education. In particular, women with higher education levels were less likely to have untreated dental caries. This finding was in agreement with Vergnes et al. [27] who showed a negative association between education level of pregnant women in France and prevalence of dental caries. Lower education level as a risk indicator for dental caries has also been described in studies among other population groups [29]. The positive relationship between education and health can be attributable to increased health knowledge and healthy behaviors, better employment opportunities and higher income, and reduced stress due to high sense of control, social standing, and social support [30].

None of the pregnant women in this study had healthy gingiva. Similarly, 100% of pregnant women examined by Acharya and Bhat [23] in India and Bressane et al. [4] in Brazil had some degree of gingival inflammation. Studies by other authors also reported high prevalence of gingivitis in pregnant women: 97.9% in Lithuania [6], 93.8% in Argentina [24], and 86.2% in Thailand [5]. The mean GI score of women in this study was 1.2 (SD 0.58) which corresponds to the finding that most of them had moderate gingivitis.

In this study, household income and perceived bleeding gums were associated with the prevalence of moderate to severe gingivitis. Bleeding gums is a common sign of gingivitis, the earliest stage of periodontal disease. About a quarter of pregnant women in this study reported having the problem. Bleeding gums, either spontaneously or during tooth brushing, was also the most commonly reported periodontal symptom among pregnant women in other population groups such as Uganda (49.8%), Spain (62.3%), and Australia (60.0%) [7,31,32]. Differences in the distribution and severity of gingivitis as well as its symptoms according to income category have also been observed in other studies that showed higher disease tendencies in lower income groups [33,34]. Socio-economic status is an important predictor for oral health attitude and behavior including dental attendance [35]. It is possible that the significant socio-economic gradient in the severity of gingivitis among women in this study was attributable to the women’s poor oral health attitude and behavior that puts them at higher risk for the problem.

Most pregnant women in this study have signs of periodontitis. The percentage of pregnant women having periodontal pockets was fairly high at 46.3%. In rural India, the prevalence of periodontal pockets among pregnant women was 33.2% [23]. In Brazil, two different studies conducted among pregnant women who received antenatal care in two different cities, Campinas [36] and Lucas do Rio Verde [25] reported distinct results, 47% and 1.1%, respectively. In Uganda, only 0.6% of pregnant women have periodontal pockets [32]. Besides the influence of social determinants, large variation in prevalence of periodontal pockets among pregnant women in different population groups is possibly because the women were at different stages of pregnancy. Progressive increase in sex hormones during pregnancy are known to be responsible for modifying periodontal tissue responses to microbial plaque that results in deterioration of periodontal health as the pregnancy progresses [37]. In agreement, in this study, the prevalence of periodontal pockets was higher among women in the third trimester than those in the first and second trimester.

Our findings also showed that women with periodontal pockets were more likely to have bleeding gums. Another significant correlate of periodontal pocket was bad breath, a common symptom of periodontitis which was the most common periodontal symptom reported by women in this study. On the other hand, women who complained of swollen gums were less likely to have periodontal pockets than those who did not have the problem. Gum swelling is a common symptom of gingivitis. Although gingivitis is a necessary precursor to periodontitis, the classic signs of inflammation like swollen gums may become less evident as the disease progress to periodontitis and gingival tissues may appear almost normal [38].

There is often a discrepancy between a person’s perceived oral health status and a professional’s assessment found in clinical examination which is the source of many misunderstandings between patients and health professionals [10,11,12,13]. A cursory look at the differences between the proportion of women in this study who reported having oral health problems to the prevalence of diseases found in clinical examination also indicates some discrepancies. For instance, toothache, which is the common symptom of dental caries, was reported by only less than a quarter of the women although more than half of them had untreated dental caries. Additionally, the common symptom of periodontal diseases such as gum bleeding was reported by only about a quarter of the women although none of them had healthy gingiva and almost half were affected by periodontitis. Further multivariable analysis, however, revealed that self-perceived oral health status was significantly associated with clinical oral health status of pregnant women in this study. In the absence of a conclusive evidence on the associations between self-reported oral health status and clinical oral health status, our findings add to the body of evidence regarding the validity of self-perceived oral symptoms.

This study has several limitations. Information obtained through self-administered questionnaire should be interpreted with caution due potential bias created through favourable responses. Sensitive questions considered by many such as income can also lead to error in reporting, creating an additional source of unreliability. In addition, most respondents in this study were Malays. Results of this study thus may not be inferred to pregnant women from other ethnic groups.

## 5. Conclusions

Our study shows that the oral health status of pregnant women who received antenatal care at Hospital USM, Kelantan was poor. The prevalence and severity of dental caries and periodontal disease were high, and these were significantly associated with the corresponding oral symptoms. Our findings highlight the potential use of an oral symptoms checklist that can be easily implemented in a busy clinical setting to estimate the oral health status and needs of pregnant women. Our study also showed that the number of women reporting oral symptoms was lower than the prevalence of corresponding oral diseases found by clinical examination. This discrepancy may be because dental caries and gingivitis are often asymptomatic at the early stages and hence may go unnoticed. These findings underline the need to educate pregnant women on the symptoms and signs of common oral diseases as this may subsequently improve the validity of self-reported tools.

## Figures and Tables

**Table 1 ijerph-17-07337-t001:** Socio-economic and obstetric profile.

Variable	Frequency (%)
Age group (years)	
≤19	5 (2.6)
20–24	22 (11.5)
25–29	66 (34.4)
30–34	61 (31.8)
35–39	26 (13.5)
≥40	12 (6.3)
Ethnic group	
Malay	185 (96.4)
Others	7 (3.6)
Highest educational level	
Primary/Secondary	61 (31.8)
Post-secondary/Diploma	70 (36.5)
Tertiary	61 (31.8)
Employment status	
No	42 (21.9)
Yes	150 (78.1)
Household income (MYR)	
<1000	37 (19.3)
1000–3000	81 (42.2)
3001–5000	50 (26.0)
>5000	24 (12.5)
Period of gestation	
First/Second trimester	96 (50.0)
Third trimester	96 (50.0)
Gravida status	
Primigravida	52 (27.1)
Multigravida	140 (72.9)
Parity status	
Primiparous	59 (30.7)
Multiparous	133 (69.3)

**Table 2 ijerph-17-07337-t002:** Dental caries experience, gingival health and periodontal health.

Variable	Frequency (%)	Mean (SD)
DMFT	179 (93.2)	5.9 (3.97)
DT	113 (58.9)	1.5 (1.75)
MT	115 (59.9)	1.7 (2.06)
FT	146 (76.0)	2.8 (2.55)
GI	-	1.2 (0.58)
0.0 (Healthy)	0 (0.0)	-
0.1–1.0 (Mild)	89 (46.4)	-
1.1–2.0 (Moderate)	90 (46.9)	-
2.1–3.0 (Severe)	13 (6.8)	-
CPI score	-	-
0 (Healthy)	2 (1.0)	1.2 (1.46)
1 (Bleeding after probing)	22 (11.5)	1.9 (1.45)
2 (Calculus)	79 (41.1)	1.9 (1.52)
3 (Pocket 4–5 mm)	88 (45.8)	1.0 (1.40)
4 (Pocket ≥ 6 mm)	1(0.5)	0.01 (0.07)

Abbreviations: DMFT = Decayed, Missing, and Filled Teeth, DT = Decayed teeth, MT = Missing Teeth, FT = Filed teeth, GI = Gingiva Index, CPI = Community Periodontal Index.

**Table 3 ijerph-17-07337-t003:** Factors associated with untreated dental caries using multiple logistic regression.

Variable	Adjusted OR	95% CI	χ^2^ Statistic (d.f.) ^a^	*p*-Value ^a^
Highest educational level	−	−	21.61 (2)	<0.001
Primary/Secondary	1.00	−	−	−
Post-secondary/Diploma	0.26	0.11, 0.61	9.76 (1) ^b^	0.002 ^b^
Tertiary	0.06	0.06, 0.36	18.26 (1) ^b^	<0.001 ^b^
Perceived cavitated tooth	−	−	−	−
No	1.00	−	−	−
Yes	4.28	2.20, 8.33	19.63 (1)	<0.001

Abbreviations: OR = Odds Ratio, CI = Confidence Interval, d.f. = degree of freedom; ^a^ Likelihood Ratio (LR) test. ^b^ Wald test.

**Table 4 ijerph-17-07337-t004:** Factors associated with moderate to severe gingivitis using multiple logistic regression.

Variable	Adjusted OR	95% CI	χ^2^ Statistic (d.f.) ^a^	*p-*Value ^a^
Household income (MYR)	−	−	−	−
≤3000	1.00	−	−	−
>3000	0.29	0.15, 0.54	15.92 (1)	<0.001
Perceived gum bleeding	−	−	−	−
No	1.00	−	−	−
Yes	2.99	1.48, 6.05	9.90 (1)	0.002

Abbreviations: OR = Odds Ratio, CI = Confidence Interval, d.f. = degree of freedom; ^a^ Likelihood Ratio (LR) test.

**Table 5 ijerph-17-07337-t005:** Factors associated with periodontal pockets using multiple logistic regression.

Variable	Adjusted OR	95% CI	χ^2^ Statistic (d.f.) ^a^	*p-*Value ^a^
Period of gestation	−	−	−	−
First/Second trimester	1.00	−	−	−
Third trimester	1.82	1.05, 3.31	3.86 (1)	0.044
Perceived gum swelling	−	−	−	−
No	1.00	−	−	−
Yes	0.29	0.10, 0.83	5.74 (1)	0.017
Perceived gum bleeding	−	−	−	−
No	1.00	−	−	−
Yes	2.16	1.08, 4.32	4.89 (1)	0.027
Perceived bad breath	−	−	−	−
No	1.00	−	−	−
Yes	1.80	1.03, 3.31	3.85 (1)	0.045

Abbreviations: OR = Odds Ratio, CI = Confidence Interval, d.f. = degree of freedom; ^a^ Likelihood Ratio (LR) test.

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
