# Peer review of "Association between Perceived Oral Symptoms and Presence of Clinically Diagnosed Oral Diseases in a Sample of Pregnant Women in Malaysia"

_ijerph, 2020, doi:10.3390/ijerph17197337_

Round 1

Reviewer 1 Report

This study aimed to determine the association between self-perceived oral health problems and professionally diagnosed oral health status in pregnant women. this is a timely and generally well-written paper. There are a number of questions that arise in the reading of this paper that are reported below.

1/ Pregnant women with diabetes mellitus or with acquired endocardial defects were not included but what about the many other chronic diseases?

2/ Could you tell us how you determined the sample size for the study?

3/ A structured self-administered questionnaire was used to obtain information. It would be interesting to present in a table the questions asked.

4/ How were the questions selected? With a steering committee?

5/ Why not use an oral health related quality of life self-assessment tool such as the Global Oral Health Assessment Index (GOHAI) or the Oral health Impact Profile (OHIP) or other…

6/ It would be interesting to focus your discussion more on perceived health. There is often a discrepancy between a person's perceived health and a health professional's clinical assessment which is the source of many misunderstandings between patient and health professional. This should be emphasized in your discussion.

Reviewer 2 Report

Review

Review:

The manuscript is interesting, and this review tries to improve this manuscript. Some suggestions may be indicated.

  1. The participation rate may be indicated. In the study the sample size is 192 women. More explicitly, what number of women did not could participate by different reasons? Or all selected women in the sampling procedure accepted to participate (100% participation rate).

  1. In the sampling procedure may be convenient to indicate the confidence level and the error alpha of the sampling.

  1. To better understand the study, an indication of the character (public or private) of dental antenatal care at School of Dental Sciences may be useful.    

  1. Add references of the oral health index: Decayed, Missing, and Filled Teeth (DMFT) Index, the Löe and Silness Gingival Index (GI), the Community Periodontal Index (CPI), may be useful for the readers.

  1. There is no mention of the study’s limitations. For example, what about the external validity of the results considering the sample size. In other words, could be the results generalized the pregnant women in Malaysia?   

  1. An alternative to the variable selection using backward in the logistic regression, is the direct acyclic diagraph (DAGs) (1), which permits to adjust for potential confounders based on relationship of exposure-outcome. Today a guidance of this approach has been published (2).  

References

1.Textor J, Van der Zander B, Gilthorpe MK, Liskiewicz M, Ellison GTH. Robust causal inference using directed acyclic graphs: The R package ‘dagitty’. Int. J. Epidemiol. 2016, 45, 1887–1894.

2.Lederer DJ, Bell SC, Branson RD, Chalmers JD, Marshall R, Maslove DM, Ost DE, Punjabi NM, Schatz M, Smyth AR, et al. Control of confounding and reporting of results in causal inference studies. Guidance for authors from editors of respiratory, sleep, and critical care journals. Ann Am Thorac Soc. 2019:16:22-28.

Round 2

Reviewer 1 Report

The authors followed all my suggestions. Therefore, I am favourable for the publication of the paper.